# Two centuries of forest succession, and 30 years of vegetation changes in permanent plots in an inland sand dune area, The Netherlands

**Karel Prach** [1,2]*, **Karol Ujházy** [3], **Vlastimil Knopp** [3], **Josef Fanta** [4]

**1** Department of Botany, Faculty of Science, University of South Bohemia, České Budějovice, Czech Republic-EU, **2** Institute of Botany CAS, Třeboň, Czech Republic-EU, **3** Faculty of Forestry, Technical University in Zvolen, Zvolen, Slovakia-EU, **4** Ke Králům 1109, Dobřichovice, Czech Republic-EU

* prach@prf.jcu.cz

## Abstract

There are not many sites in densely populated temperate Europe where primary forest succession has a chance to run without direct human intervention for a long time and over a relatively large area. The extensive drift sand area of the Veluwe, central Netherlands, provided an opportunity to study succession in a formerly open and dynamic inland sand dune system combining chronosequence and permanent plot approaches. Different successional stages, aged up to 205 years since the first tree individuals established, were identified and vegetation studied using 1200 permanent plots established in 1988 in three adjacent sand dune complexes of different successional age, and resampled during the past three decades. After two centuries, forest succession has proceeded to a pine forest with gradually increasing participation of native deciduous trees. However, their expansion has been arrested by browsing of wild ungulates. Species diversity peaked after about 40 years of forest succession, then declined, and increased again after 100 years. During the past three decades, the herb layer has differentiated in the oldest plots, and the spontaneous forest succession is still in progress. Besides open drift sand with early successional stages, also the spontaneously established late successional forests are valuable from the conservation point of view.

## Introduction

Dunes cover about 7% of the terrestrial surface of the Earth but only a small portion is still active. Natural stabilization of dunes can be caused both by abiotic (geomorphologic barriers, changes in wind directions) and biotic factors (ongoing succession). Both types of factors usually operate in concert [1]. Moreover, humans have intentionally stabilized many dunes, either by creating artificial barriers or by planting or seeding mostly trees and shrubs to speed up natural stabilization [2]. When dunes become stable, succession usually accelerates, but some dunes may remain unstable for thousands of years.

**Data Availability Statement:** Data are available in Zenodo: 10.5281/zenodo.4680732.

**Funding:** The research was supported by the Slovak Grant Agency VEGA (projects no. 1/0639/17

and 1/0624/21), and by the project no. 20-06065S granted by the Grant Agency of the Czech Republic. The funders had no role in study design, data collection and analysis, decision to publish, or preparation of the manuscript.

**Competing interests:** The authors have declared that no competing interests exist.

Dunes have been central to the development of early ideas of succession [3, 4]. Since that time, various sand dunes have been studied in many parts of the world, belonging to the best examples of spontaneous primary succession (see summarised information in [5–9]). The majority of studies has dealt with succession in coastal dunes [5]. An overview of the development of NW European coastal dunes and their vegetation was already published [10]. However, many studies of dunes do not directly incorporate succession. They rely on a toposequence approach, where physical features (dune swales, dune ridges, distance from shoreline) become substitutes for time [11]. Resulting insights must be used with caution because often only weak relations to real successional age exist [9]. In sand dune systems, succession usually starts in different parts at different times owing to differential dune stabilization and other topographic factors [12]. Succession is usually divergent, or multiple successional pathways are followed [13] but not always. In contrast to most studies, we earlier found a rather uniform course of forest succession over a nearly entire inland sand dune system [14, 15]. In temperate Europe, succession on sand dunes usually proceeds to woodland. Occasionally, invasive alien species may establish and then some measures controlling them may be desired [5, 9].

There are not many sites in densely populated temperate Europe where primary succession has a chance to run nearly without human intervention for a long time and over a relatively large area. The extensive drift sand area of the Veluwe in central Netherlands provided an opportunity to study succession from a formerly open and active inland sand dune system to a Scots pine (*Pinus sylvestris*) forest [14, 15]. In the Netherlands, with extensive strips of coastal dunes, attention used to be mostly directed to these [13, 16–18] while inland dunes received less interest in the past. Earlier, the prevailing effort in various parts of Europe was to stop sand dune movement to protect arable land, pastures, roads and settlements [19]. Extensive artificial pine afforestation accelerated in the 19th century in the studied area and elsewhere in NW Europe [20]. Currently, after the extent of active sand-blown areas has largely diminished [21], nature conservation often tries to stop further stabilization of sand dunes or even to restore their active movement with various forms of restoration management [22]. Also elsewhere in Europe, some inland dune ecosystems are actively being restored for their specialized flora and fauna [23, 24]. There are nowadays especially attempts to maintain high-biodiversity open habitats requiring continued, intense human intervention in the landscape by artificial mechanical disturbances, reintroduction of grazing, and reducing nitrogen and organic matter levels (e.g. [25]). This is also the case in the studied area, with the last sites of sand-blown dunes, which used to be rather limited in extent but later slightly restored by mechanical disturbance. However, a more advanced forest succession means more effort to restore active sand dunes. The habitat of open sand represents one of the endangered Natura 2000 habitats [26], but also spontaneously developed, late successional sand dune forests may have conservation value and may represent a more natural way of afforestation than the artificial planting which is still being practiced in various sand dune areas [27]. Consequently, we also attempted to formulate some conclusions which can be potentially used for nature conservation and forest restoration.

We asked the following main questions: (a) What is the course of forest succession over the two centuries? (b) How has the forest vegetation in permanent plots changed in the past three decades? (c) How has the species diversity changed during the course of forest succession? (d) How much do native deciduous woody species establish in successional pine forests?

## Material and methods

### Study sites

The plots studied were located in the Hulshorsterzand and the Leuvenumse Bos Nature Reserves owned and managed by Natuurmonumenten, a Dutch NGO for nature conservation,

near the town of Harderwijk, in the northern part of the Veluwe region in central Netherlands at an altitude between 9 and 24 m a.s.l. Natuurmonumenten kindly agreed with our research. The climate is temperate humid. The mean annual temperature is 9.4°C, and the annual rainfall is about 820 mm [28].

The entire region was markedly affected by fluvial and glacial processes in the past (details described in [12]). During the Holocene period, the whole landscape became covered by closed-canopy forest, and deep podzol soils developed. Mixed forests with *Quercus robur*, *Betula* spp. and *Pinus sylvestris* dominated the area; *Fagus sylvatica* became a dominant species, at least locally, later in the Holocene [29, 30]. The forests started to be largely influenced by human activities, especially by grazing of domestic animals and cutting, as early as about 2800 BC. Heathland is reported to have been present over a large area around 700 AD, replacing the declining forest. In the Middle Ages, after 1250 AD, the heathland was mostly destroyed by overgrazing, and the sand became active again. In the first half of the 19[th] century, intensive sheep grazing started to decrease and the forest gradually expanded by both spontaneous establishment and planting (for details on the history of the area, see [2, 12, 31]).

Different successional stages of up to 175 years since dune stabilisation, indicated by establishment of pines, were identified there in the late 1980s [14, 15]. Succession in the inland dunes in the Veluwe region was preliminarily described by [14], who showed the main trends in vegetation and soil development. Vegetation was assessed in detail by [15]. Some vegetation analyses were repeated after 15 [32, 33] and 30 years (this study). Changes in soil development, and effects of soil characteristics, organic matter and nutrient fixation on plants during primary succession were described in details by [34].

## Field methods

Three large adjacent permanent plots (A, B, C) of 200 × 200 m in size, were established in 1988. One was located in one of the remaining partly active sand-blown areas (A), while the remaining two were established in two differently aged closed forests covering former sand-blown sites (B and C). Coordinates of the centres of the research plots are: A– 52°20'43.5"N, 5°43'41.9"E; B– 52°20'38.2"N, 5°43'58.8"E; C– 52°19'26.9"N, 5°41'48.6"E. Vegetation in the plots established spontaneously and has not been evidently influenced directly by human activities. We have also tried to include into the plots proportionally all the geomorphological units distinguished, and to avoid sites of uncertain history. The size of 200 × 200 m was sufficient to encompass the diverse geomorphology [2]. Each plot was permanently marked at 50 m intervals by means of concrete pillars and buried metal marks. Each of the plots was subdivided into a grid consisting of 10 × 10 m subplots, resulting into 400 subplots per plot. All the subplots were sampled by making phytosociological relevés in 1988 (in all three large plots), 2003 (only herb layer) and 2018 (only plot C because the other ones had been partly disturbed in the meantime). The cover of all vascular plants, the dominant moss, characteristic for sandy substrates, *Polytrichum piliferum*, and the cover of bare substrate and litter were estimated using an ordinal scale 1–9 [35]. The cover of herb, shrub, and tree layers was visually estimated in percentage. The following mapping units were *a priori* distinguished based on dominant species of the herb layer, and *a posteriori* fully confirmed by the TWINSPAN analysis [33]: 1. *Ammophila arenaria* and/or *Festuca arenaria*; 2. *Corynephorus canescens*, locally accompanied by *Polytrichum piliferum*; 3. *Festuca ovina* locally accompanied by *Agrostis vinealis;* 4. *Deschampsia flexuosa*; 5. *Empetrum nigrum*; 6. *Vaccinium myrtillus*; 7. *V. vitis-idaea*; 8. Other species with low occurrence (*Calluna vulgaris*, *Molinia caerulea*); 9. No vegetation (bare sand or locally tree litter). The units were mapped in each subplot, then vegetation maps of the entire plots were constructed [33].

The following relief types (geomorphological units) were mapped in each subplot (abbreviations used hereafter are in brackets): Plain (P)–flat area with fluvio-glacial deposits where drift sand was blown out and fine-sized gravel remained; Low dunes (LD)–small accumulations of sand up to ca 1.5 m high; High and steep dunes (HD); Plateau dunes (PD) with the fossil horizon at or near the surface; and Wet-level areas (WL) with a high ground water table [15].

The age of selected (107) individuals of *Pinus sylvestris* throughout the plots was estimated using a Pressler wood-core auger in 1988. The cores were taken as close to the ground as possible, and the number of tree rings counted used as an environmental variable closely related to length of succession (Age). In each distinct patch obviously homogenous in its vegetation structure, thus of the same expected age, several of the probably oldest specimens were probed. If no tree was cored in a particular subplot, the average tree age in the patch was assigned to the subplot. The age range of the sampled pine trees was as follows (data related to 1988, when the trees were sampled): Plot A 1–40 years; Plot B 30–112; Plot C 110–175. This means that the oldest subplots analysed in 2018 were 205 years old. For details on methods, see [32, 33]. (*Note*: it must be emphasised that the age of pines cannot be directly considered as equal to successional age because herbal vegetation usually established prior to the pines and obviously existed for a very variable length of time, which becomes shorter with decreasing extent of the active sand-blown area.)

## Data analysis

Vegetation relevés were entered into the TurboVeg database [36] and processed by the JUICE programme including previous transformation of the ordinal values into mean respective cover values [37]. Average percentage cover values (including zero values–Barkman's TCV calculated in the JUICE programme [37]) from all 400 relevés, and frequencies, based on the number of subplots in which the species occurred.

Detrended Correspondence Analysis (DCA) was applied using CANOCO 5.0 [38] after logarithmic transformation of the cover values. The length of the gradient was 4.4 SD units. A matrix of 1600 samples (10 × 10 m) and 59 species (58 taxa of vascular plants and *Polytrichum piliferum*) was used for the ordination analysis. Woody species occurring in multiple layers were combined into a single one in the JUICE programme. Environmental factors (successional age, relief types) and some community properties (vascular plant species number–SR, Shannon-Wiener species diversity–H') were used as additionally plotted passive variables [39]. Percentage proportion of each relief type and percentage of bare sand within a subplot were set as numeric environmental variables. Pearson coefficients of linear correlation between environmental variables and the first axis scores of the DCA as well as their significances were calculated in the Statistica ® programme.

Direct influence of environmental variables (year of sampling, successional forest age, relief type and their interactions), on the herb layer composition was tested using a set of 400 subplots of plot C (resampled 1988, 2003, 2018) with the Monte-Carlo permutation test within the partial Redundancy Analysis (RDA) ordination. The linear type of gradient analysis was used since the gradient length was 2.9 SD units. Permutations were restricted according to the following sampling design: i) for the rectangular grid (400 subplots) to avoid effects of spatial autocorrelation, and ii) for assignment of the resampled subplot (subplot number was set as covariable).

Species response curves for the main woody species were also calculated in CANOCO using Generalized additive models (GAM) based on use of a smooth semi-parametric term [38]. They were produced separately for their participation (percentage cover in the subplots) in the herb layer (seedlings or saplings up to 0.5 m in height) and in the combined shrub and

tree layer (the combined cover values being calculated following [40]) using plots and years when all layers were sampled (A 1988, B 1988, C 1988 and C 2018).

Species diversity (the Shannon-Wiener index) in each of the subplots was plotted against successional age, and significance of the relationship fitted by a third degree (cubic) polynomial function was calculated in the Statistica ® programme. The data based on mapping, i.e. the pattern of vegetation units over the plots A, B and C, were vectorised in ArcGIS (ESRI, Redlands, CA). Because vegetation maps of plots A and B were already published earlier [32, 33], we only present here the maps from the recently (2018) repeated plot C to visualize successional changes in the late successional forest stands. However, the total areas covered by particular mapping units (in %) in the respective sampling years were calculated for all the plots (A, B and C).

## Results

Altogether, 65 taxa of vascular plants including 22 woody species were recorded in the studied succession. Six of the woody species were alien species, three of which reached some abundance: *Pseudotsuga menziesii* and *Quercus rubra* with negligible cover, more frequently *Prunus serotina* with low cover values locally exceeding 5%. Three other aliens occurred only very incidentally (see S1 Table). The cover values and frequencies of each recorded species in plots A, B and C are presented in the summarizing S1 Table. The primary data are deposited in Zenodo (DOI: 10.1371/journal.pone.0250003).

The successional changes during the whole considered period of ca 200 years, based on the complete set of vegetation relevés, are clearly visible in the ordination diagram presented in Fig 1 (see also Figs 2–4). The first ordination (DCA) axis well reflects successional age (correlation between sample scores and age: $r = 0.92$, $p < 0.001$) and is negatively correlated with bare sand proportion which reached the highest percentages in the early successional stages of plot A ($r = -0.88$; $p < 0.001$). Correlations with other geomorphological units were low. Species richness and diversity increased with age, i.e., increased along the first axis. The differences between the plots A, B and C are clear, as well as the shift of plot C between the sampling years of 1988 and 2018. The succession seems to be divergent at the beginning, when high dunes and plains differ in colonisation by the first species (*Ammophila arenaria* and *Festuca arenaria* vs. *Polytrichum piliferum*, *Festuca ovina* and *Agrostis vinealis*)–see the passively projected variables of the relief types. Then the succession appeared to be convergent when a closed-canopy pine forest established while bare sand gradually decreased as succession advanced. The ordination of species (those with the highest weight are displayed) well reflects a successional sequence of dominants and co-dominants during succession. It started with *Ammophila arenaria* occurring on high dunes, being followed by *Corynephorus canescens* dominating nearly everywhere, and the already mentioned *Polytrichum piliferum*, *Festuca ovina*, and *Agrostis vinealis* prevailing especially on plains. Establishment of *P. sylvestris* is a crucial point in succession. This species has a central role and kept its position until the end of the observed period, being accompanied by some other woody species. *Fagus sylvatica* appeared to be the most successful among the native deciduous trees. In the herb layer, *Deschampsia flexuosa* dominated at first, being gradually replaced by *Empetrum nigrum*, *Vaccinium myrtillus*, and *V. vitis-idaea*. (See also Supporting Information).

In the partial RDA ordination, the Monte-Carlo permutation test demonstrated highly significant influence of year of sampling, age of the trees, relief types (especially plateau dunes and plains) and their interactions with age (S2 Table) on the overall vegetation composition. The relief types, used as passive variables in the DCA and plotted in the ordination diagram (Fig 1), indicate that high dunes were important in the initial successional stages, whereas plateau dunes and plains caused differentiation of species composition in later development.

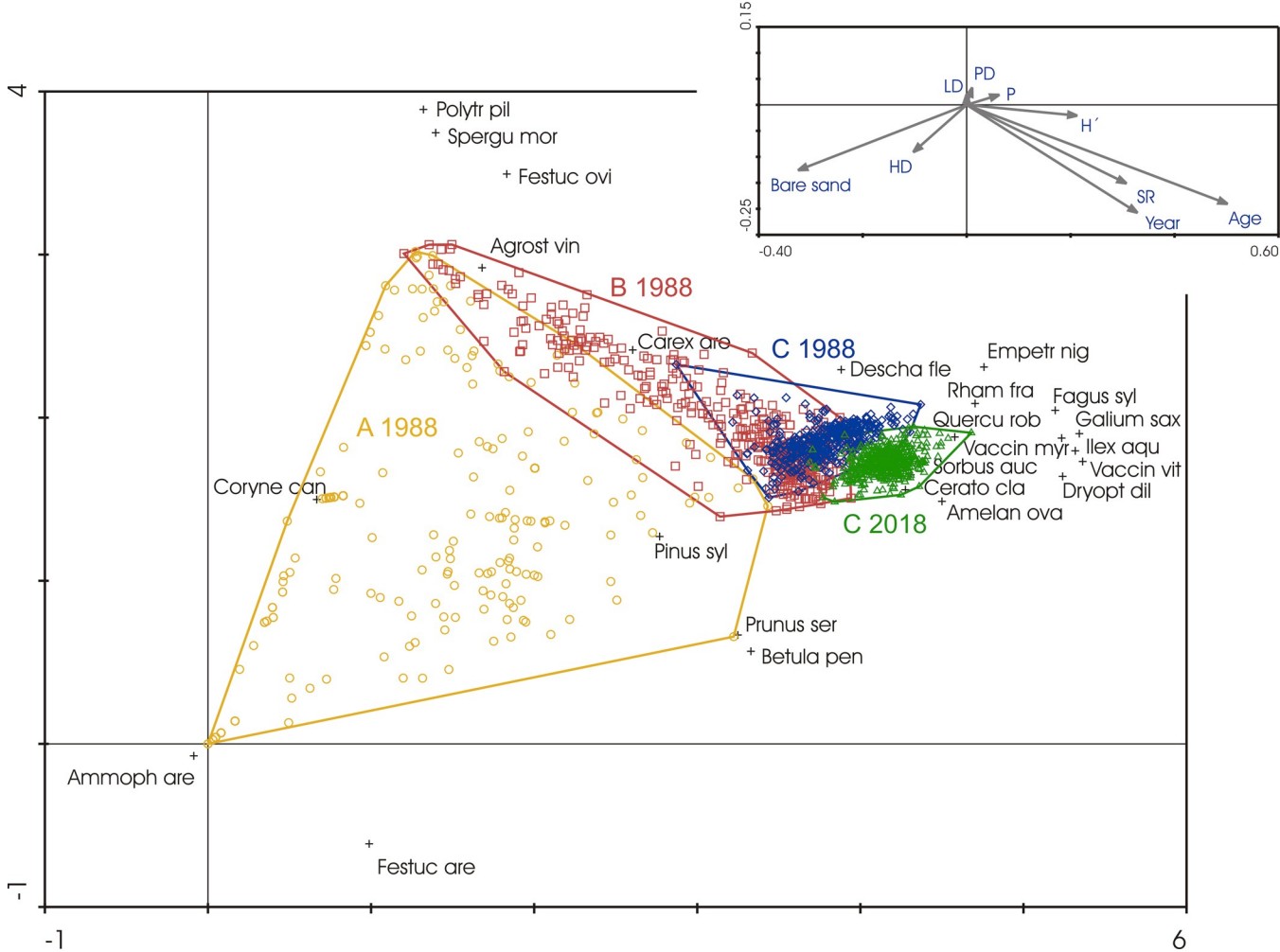

**Fig 1.** Biplot of DCA with samples classified (and outlined by envelopes) according to plot assignment (A, B or C) and year of sampling, and species with the highest weight (layers of woody species were combined into a single one). Percentage cover values were transformed with a logarithmic function. Eigenvalues of the first two axes were 0.727 and 0.211; total inertia 4.789. Vectors of supplementary variables are shown in the upper right corner: P–Plain; LD–Low dunes; HD–High dunes; PD–Plateau dunes; SR–Species richness; H'–Species diversity of vascular plants; Age–successional age; Year–year of sampling.

Changes in species diversity are presented in Fig 5. It generally increased (see also the passively projected community variable in Fig 1). After the moment when a close pine stand established, there was a slight decline culminating at about 100 years after the establishment of the first pines, i.e. in the time of high dominance of *Deschampsia flexuosa*. Later, the diversity increased again, as other woody and herb species increased in abundance (Fig 5). In a detailed view, the average number of species per subplot changed as follows: Plot A, 1988–2.3; Plot B, 1988–7.1; Plot C, 1988–5.5; 2003–7.4; 2018–10.8. The average number of species in Plot C doubled during the past three decades. Small-scale diversification of the herb layer vegetation in later successional stages in Plot C is evident from this increasing average number of species per subplot, while the total number of species did not change substantially between the years of observation: 1988–31; 2003–39; 2018–35.

The successional pattern of woody species is presented in Fig 6, separately for the combined shrub and tree layer and the herb layer, where the species occurred in the form of seedlings or small saplings. It was evident that after the strong early dominance of *Pinus sylvestris*, some

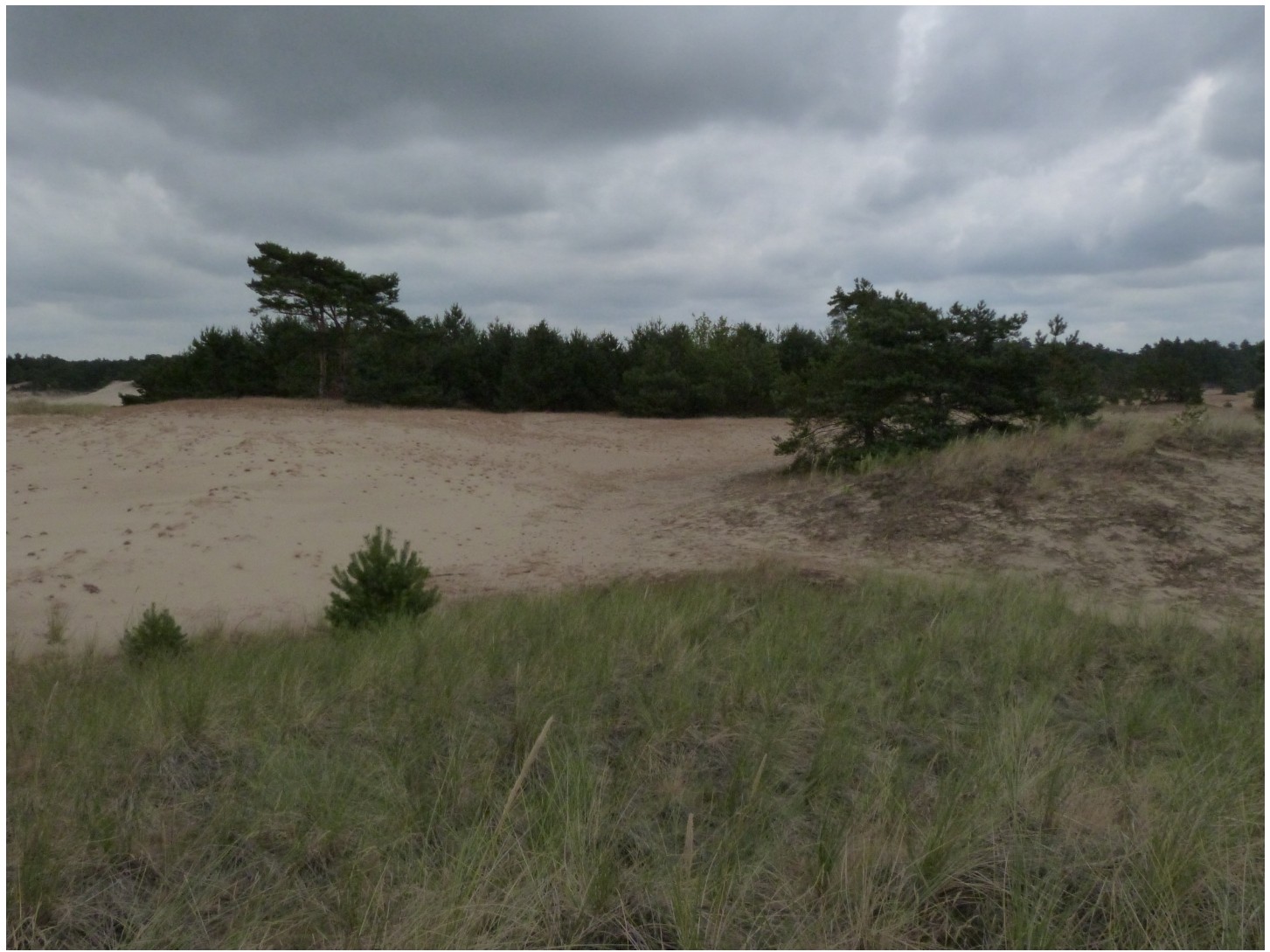

**Fig 2. Different successional stages at the youngest plot A in 2015.** Photo by K. Prach.

broadleaved trees had a chance to establish, especially *Fagus sylvatica* and probably also *Ilex aquifolium*, which is indicated by its high occurrence in the herb layer in the last year of observation (2018).

Vegetation maps of the herb layer in plot C recorded in 1988, 2003 and 2018 are presented in Fig 7. The average cover of all the mapped units over all the sampling years is summarised in Table 1, where the successional changes, corresponding to the results of the ordination described above, are clearly evident: all units representing open vegetation gradually disappeared while the units representing a typical forest herb layer established, at first dominated by *Deschampsia flexuosa*. Between the years, there is an obvious increase in the cover of the units dominated by *Vaccinium myrtillus* and *V. vitis-idaea* in the oldest plot C, while the mapping unit dominated by *Empetrum nigrum* locally fragmented in favor of the former two species, similarly as the unit dominated by *Deschampsia flexuosa*. It is obvious that the herb layer has diversified during the past three decades (Fig 7).

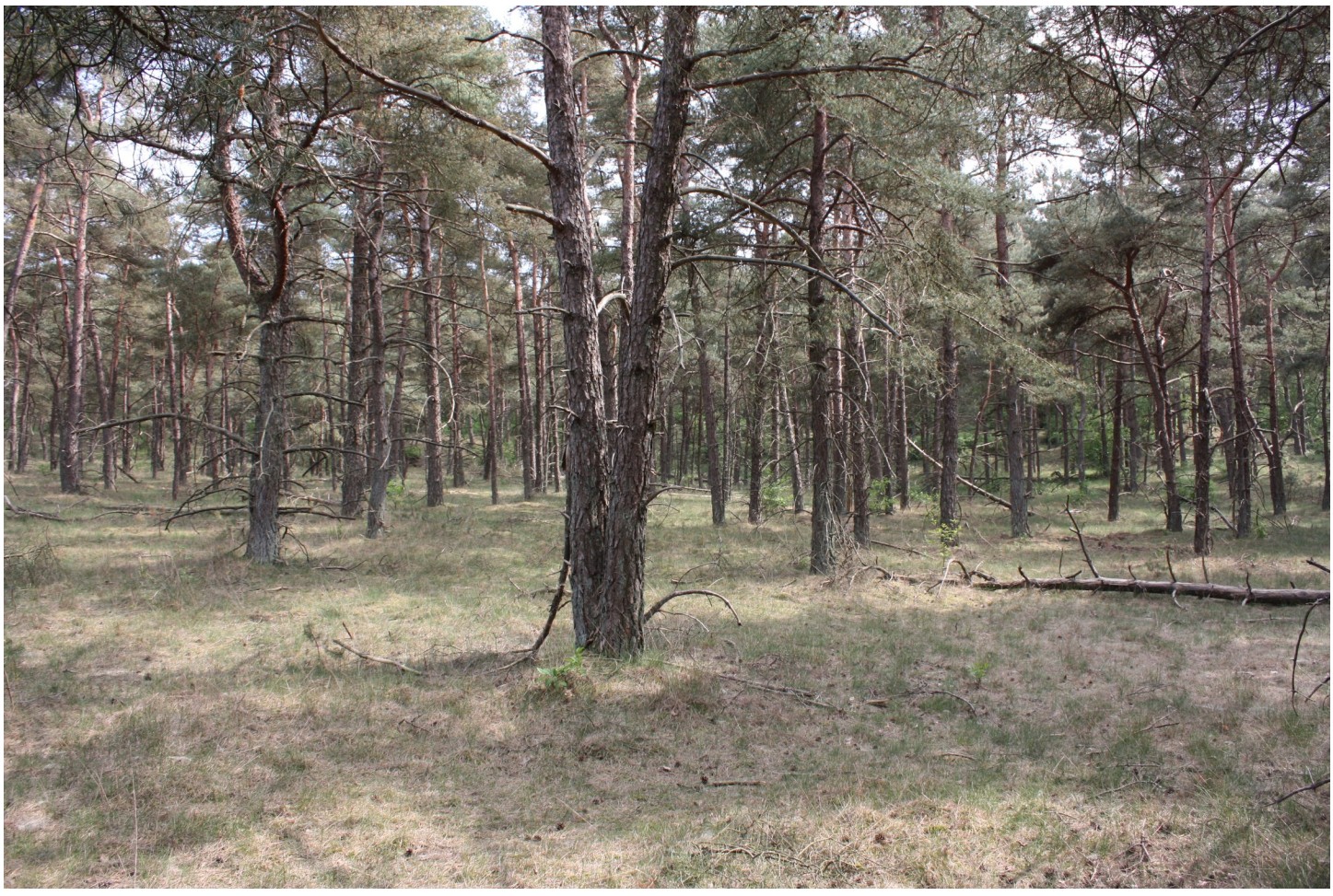

**Fig 3. A rather monotonous pine forest at plot B in 2010.** Photo by K. Ujházy.

## Discussion

During two centuries, an acidophilous pine forest with a typical herb layer [41] has developed by spontaneous succession. The frequency of late successional deciduous woody species gradually increased but only in the past several decades. However, they still reach a rather low dominance especially in the tree layer, despite their earlier abundance in the seedling stage [15]. The primary forest succession described here represents one of the oldest forest chronosequences in Europe proceeding from bare ground without any direct human interference evident. However, the earlier intentionally increasing numbers of deer and wild boar in the whole area can be regarded an indirect human influence [42]. We observed especially broadleaved woody species to be browsed in all the plots, especially their seedlings and saplings. We expect that without this impact, the pine forest, dominating in the area for two centuries, would today probably be substituted by mixed woodland with a higher dominance of broadleaves, especially by *Quercus robur* and *Fagus sylvatica*. Also other deciduous trees could potentially contribute, as is evident by their increasing presence during the second century of succession especially as seedlings and small saplings in Fig 6. Thus, wild herbivorous ungulates probably only slow the succession down towards the prevailing deciduous trees. *Fagus sylvatica* and *Quercus robur* were expected to form a terminal (or climax) woodland in this area [29], similarly to other parts of the northwestern European lowlands

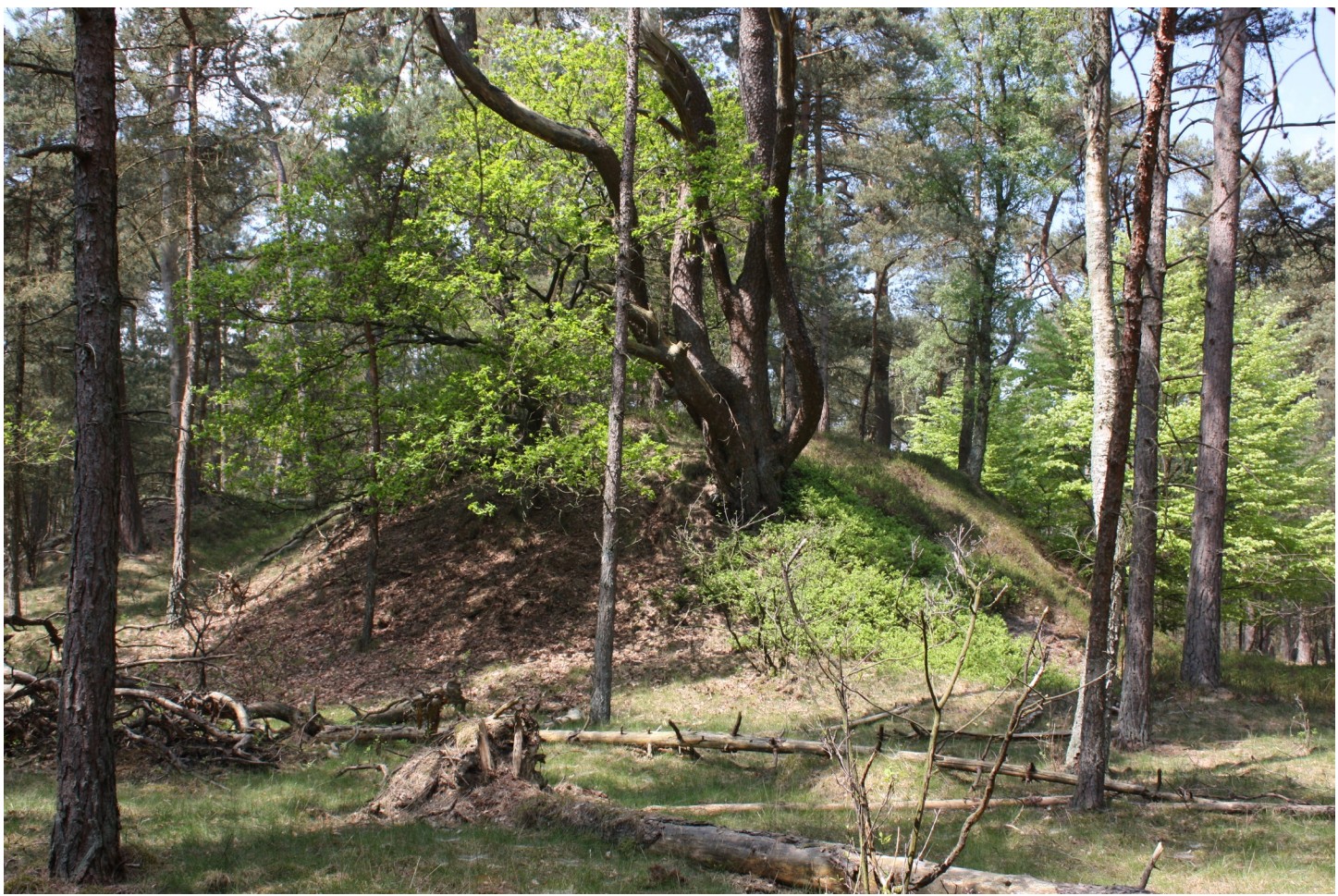

**Fig 4. Part of plot C with the oldest (ca. 200 years) pines and deciduous trees, mostly beech (*Fagus sylvatica*) in 2015.** Photo by K. Ujházy.

[30, 43, 44]. Beech seems to be more vigorous in our study plots than oak and actually dominates in remnants of natural forest in the Veluwe area. Similar evidence is reported from other regions of sandy glacial deposits of the North European Plain [45, 46], where beech has successfully spread to both pine and oak forests. The limited regeneration and competitive ability of oak compared to beech is largely a consequence of preferred browsing, as was proved by a fencing experiment in the neighboring part of the Veluwe region [47].

Dunes are considered susceptible to invasion by alien plants because of their high level of disturbance and usually open vegetation character [48, 49]. Although some woody aliens are present, higher participation and invasion are not yet evident in unmanaged parts, which is a positive message for nature conservancy. Only *Prunus serotina* seems to have a high invasion potential as reported from similar environmental conditions elsewhere [50] and may pose some threat in the future even in the area under study. However, we observed its being browsed intensively.

Early successional species are represented by sand dune specialists occurring on still unstabilized sand [51], such as *Ammophila arenaria* and *Festuca arenaria* in our case. *Corynephorus canescens* can grow in initial successional stages on both dunes and flat fluvio-glacial plains between the dunes. When the species forms some cover, there is a chance for pines to establish. However, pines can massively establish only if late spring and early summer are sufficiently

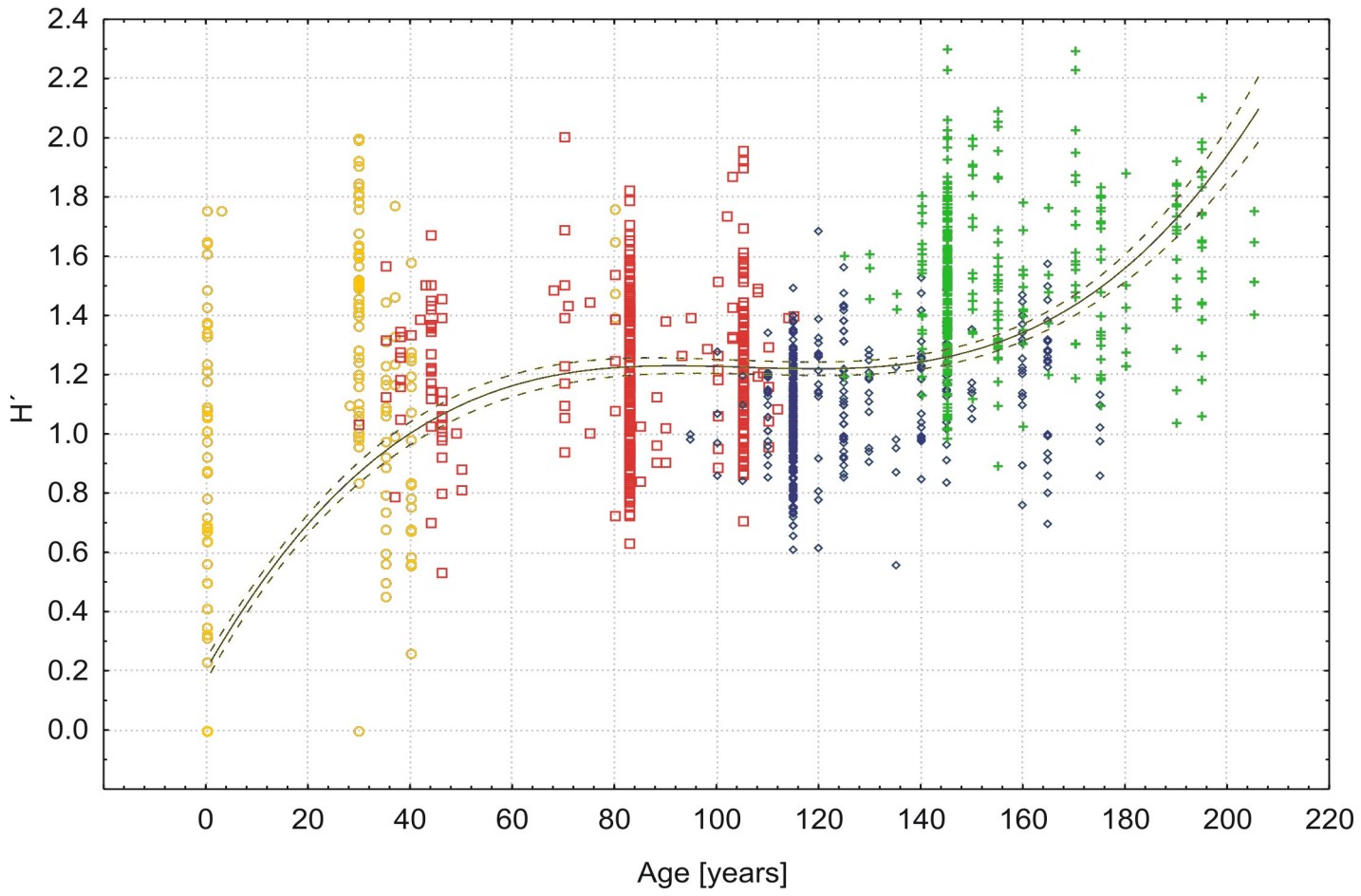

**Fig 5. Relationship between vascular plant species diversity (H': Shannon-Wiener index) and successional age in 100 m² plots.** Fitted by 3rd polynomial function (black solid line: r = 0.719; p < 0.001) with 95% confidence interval (black dashed lines). Symbol colors correspond to those used in Fig 1 for plots and years of sampling.

wet, otherwise seedlings die by drought [34]. This may explain the time gaps, evident from Fig 5, when forest stages of some ages are missing. Prior to this crucial step in sand dune succession, i.e. pine establishment, the successional development can be arrested or returned to an earlier stage by accumulation or deflation of the sand. The average time between completely bare ground and the establishment of the fist pine was roughly estimated by extrapolation to some 50 years [15] Consequently, the entire succession is approximately this time longer than quantified in this paper. Still we can speak of well-dated forest succession.

After this moment, i.e. the establishment of the first pines, succession runs towards a closed-canopy pine forest, its herb layer first dominated by *Deschampsia flexuosa*. The species dominates in a period of ca 40–130 years of forest succession, representing a uniform successional stage which masks the different starting times of succession over the relief types. Later on, regarding relief types, some differentiation can be seen in the stage of dwarf shrubs. Particularly *Empetrum nigrum* preferably expands on relatively moist plateau dunes and the northern slopes of high dunes, *Vaccinium myrtillus* is typical of mesic sites and *V. vitis-idaea* of dry fluvio-glacial plains. The rather simple direction of the succession is illustrated by the fact that it could roughly be described using only 13 properly localized phytosociological relevés in the earlier pilot study [14]. Our analysis using 2800 relevés confirmed the main earlier

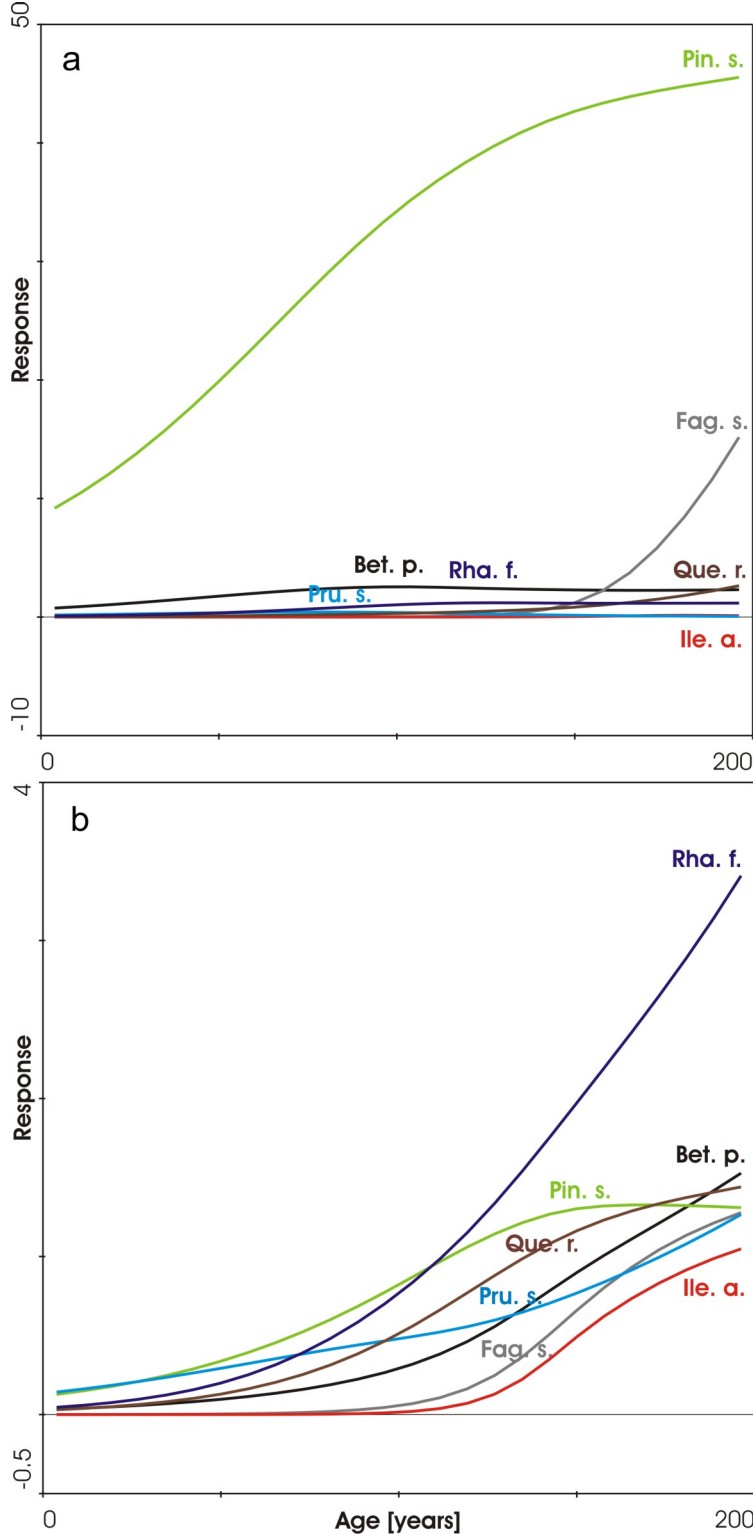

**Fig 6.** Response curves of woody species in the combined tree and shrub layer (a; up) and for woody species in the herb layer (b; down). Percentage cover values estimated in 100 m$^2$ plots were used as a response variable and the Poisson distribution was set using a GAM model. Names of species are abbreviated as follows: Bet. p.–*Betula pendula*; Fag. s.–*Fagus sylvatica*; Ile. a.–*Ilex aquifolium*; Pin. s.–*Pinus sylvestris*; Pru. s.–*Prunus serotina*; Que. r.–*Quercus robur*; Rha. f.–*Rhamnus frangula*.

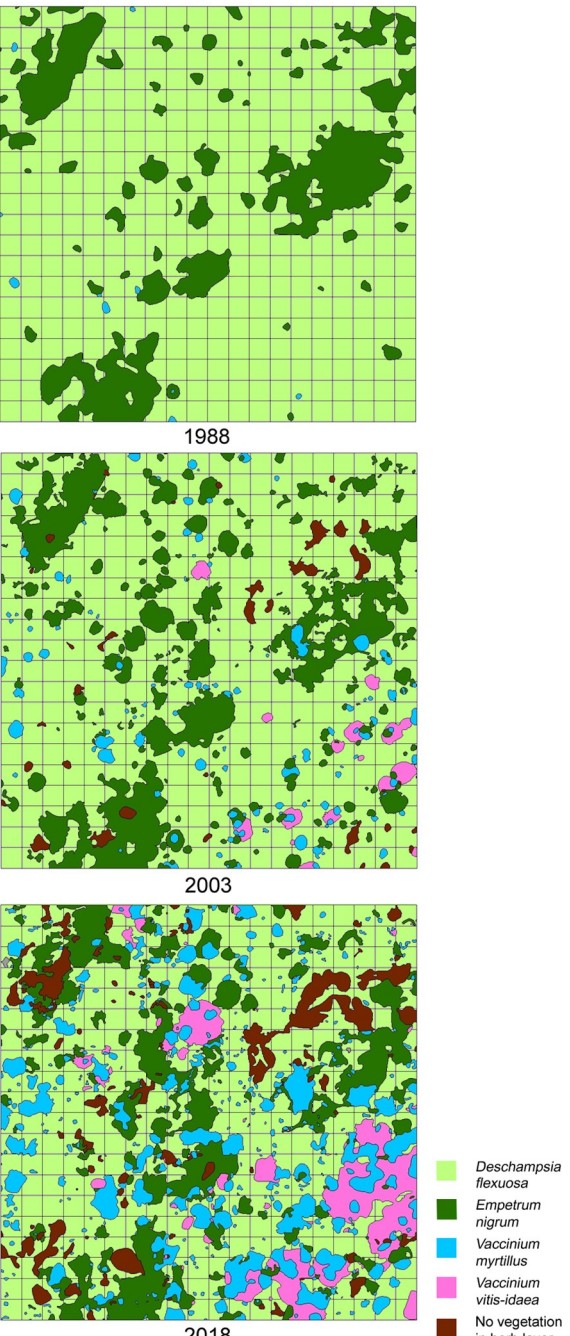

**Fig 7. Repeated vegetation maps of the successional pine forest in plot C (200 × 200 m).** Vegetation types were delimited according to the dominant species of the herb layer. The sampling grid of 10 × 10 m subplots is shown.

expectations of the course of succession. However, the rapid expansion of *Vaccinium* dwarf shrubs in pine stands older than 120 years was not predicted in the time of the first brief survey. A similar sequence of dwarf shrub expansion was described from successional forest development in abandoned heathlands in Denmark [52].

In homogenous environments, succession is mostly unidirectional, while in heterogeneous environments, multiple successional pathways are usually expected [8]. The latter were well

**Table 1. Percentages of vegetation types (mapping units) in plots (A, B, C) in sampling years (1988, 2003, 2018).** –missing data; 0.0 negligible values.

| Plot/year | A/88 | A/03 | B/88 | B/03 | C/88 | C/03 | C/18 |
|---|---|---|---|---|---|---|---|
| No vegetation | 71.5 | 44.6 | 0.0 | 0.5 | 0.0 | 2.0 | 6.9 |
| *Ammophila arenaria-Festuca arenaria* | 4.4 | 5.2 | | | | | |
| *Corynephorus canescens* | 17.6 | 38.9 | 6.1 | 2.7 | | | |
| *Festuca ovina-Agrostis vinealis* | 3.7 | 9.1 | 18.0 | 6.7 | | | |
| *Deschampsia flexuosa* | 1.5 | 0.9 | 73.7 | 83.1 | 81.4 | 71.9 | 53.2 |
| *Empetrum nigrum* | | 0.0 | 2.0 | 6.6 | 18.4 | 21.8 | 18.3 |
| *Vaccinium myrtillus* | | 0.0 | 0.0 | 0.2 | 0.2 | 2.8 | 14.7 |
| *Vaccinium vitis-idaea* | | | | | | 1.5 | 6.8 |
| Other types | 1.30 | 1.30 | 0.20 | 0.20 | 0.00 | 0.00 | 0.10 |
| Average cover of $E_1$ species | 8.5 | 26.7 | 64.3 | 84.7 | 55.3 | 75.4 | 60.6 |
| Average cover of $E_2+E_3$ species | 7.4 | 8.1 | 41.8 | – | 40.5 | – | 52.0 |

documented for various coastal sand dune systems [13]. In the studied area, despite its heterogeneity in relief forms, succession is generally unidirectional, disregarding the initial divergence prior to the massive establishment of pines, and very limited extent of sites with different site conditions (wet depressions or exposed thick fossil horizons), where succession can be somewhat modified [53]. Successional trajectories are generally expected to be convergent if the inner heterogeneity of abiotic factors is ameliorated by succeeding biota, and divergent or parallel if the initial differences persist or are even enforced during successional development [54]. In our case, succession seems to be divergent before *Pinus sylvestris* established over the whole area and after this, convergence is evident when the pine canopy is closed and *Deschampsia flexuosa* dominated the herb layer after about 100 years of forest succession [33]. Some divergence recently appeared again, especially in the herb layer, where three dwarf shrub species alternately dominated besides the locally persisting dominance of *D. flexuosa*. Consequently, this succession cannot be simply assessed as divergent or convergent.

A distinct feature of the succession described here is the variation in the colonisation of particular relief forms: first large dunes with fossil horizons, last the plains where sand had been blown out and fluvio-glacial gravel sediments remained. The time difference between the oldest and youngest subplots in plots B and C is approximately 70 years. We refer to this as asynchronous succession [33]. This described process of asynchronous colonisation of different relief types seems to be applicable to sand dune systems in general, because principally the same abiotic factors, especially wind and substrate texture, determine the formation of dunes [12], although different species may play a role. In this respect, sand dune successions are rather exceptional. In other successions, colonisation processes often start at the same time over an exposed area, although they may differ in speed and direction depending on environmental heterogeneity of the area.

Species richness and diversity usually increase in dune succession as is typical of primary successions, although they may sometimes either peak or drop in the middle of the succession. The latter happens if a species strongly dominates [9]. That is exactly the case in the succession studied here, with strong dominance by *Pinus sylvestris* in the tree layer and *Deschampsia flexuosa* in the herb layer. By this time (until about 100 years), early successional heliophilous species had already disappeared and some shade-tolerant species typical of the herb layer of late successional forests had not established yet. Afterwards, *Deschampsia flexuosa* gradually decreased and diversity again increased in our dataset which corresponded with traditional theoretical expectations [55].

There are some implications for a possible future development of the successional forests. In the case of deer and wild boar reduction, the succession would probably be accelerated

towards mixed or even deciduous woodland instead of persistence of the pine forest. Under the present game density, a slow further increase of deciduous woody species, especially beech, can be expected in the pine forest, although canopy pines can re-establish as subsequent generations, which is obvious in the understorey. Late successional forests are generally still rather rare in The Netherlands [56] and as we have reported here, they can be effectively restored by means of spontaneous succession, at least in inland sand dunes. However, since active inland sand dune areas are even rarer in NW Europe as a whole than late successional forests, the possible conflict between afforestation and maintaining or restoring the open habitat should be carefully balanced. In any case, we recommend not to perform any direct human interventions in the form of forest or nature conservation management in late successional forests and to keep the oldest existing study plot (C) as a natural laboratory. It is permanently fixed and the analyses of all the subplots can be again repeated in the future.

## Supporting information

**S1 Table. Species frequencies (Fr) and average % covers (Co) in plots A, B and C in the years of observation.** Species are sorted according to their participation in the succession. Frequencies and covers of woody species are combined in a single layer.
(DOCX)

**S2 Table. Marginal effects of environmental variables in partial RDA analysis.** All considered variables explained 23.0% of total variation (year of sampling; successional age; P–Plain; PD–Plateau dunes; LD–Low dunes; HD–High dunes). Interactions of two variables are marked with an asterisk. P values were adjusted by the Holm correction.
(DOCX)

## Acknowledgments

We are grateful to Natuurmonumenten (NL) for technical support and agreement to carry out the fieldwork in their nature reserves, and Frits Mohren for logistic support. We also thank Jan Willem Jongepier for language revision, Marek Čiliak for help with statistical analyses and reviewers for their comments.

## Author Contributions

**Conceptualization:** Karel Prach, Josef Fanta.

**Data curation:** Karol Ujházy, Vlastimil Knopp.

**Formal analysis:** Karol Ujházy, Vlastimil Knopp.

**Investigation:** Karel Prach, Karol Ujházy, Vlastimil Knopp.

**Methodology:** Karol Ujházy.

**Supervision:** Josef Fanta.

**Writing – original draft:** Karel Prach, Karol Ujházy.

**Writing – review & editing:** Karel Prach, Karol Ujházy.

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
