## [Decision Letter · Decision Letter 0]

7 Jan 2021

PONE-D-20-36698

Two centuries of forest succession, and 30 years of vegetation changes in permanent plots in an inland sand dune area, The Netherlands

PLOS ONE

Dear Dr. Prach,

Thank you for submitting your manuscript to PLOS ONE. After careful consideration, we feel that it has merit but does not fully meet PLOS ONE’s publication criteria as it currently stands. Therefore, we invite you to submit a revised version of the manuscript that addresses the points raised during the review process.

Please make revisions according to the suggestions of the reviewer,especially further clarify your process of data aquisition and methods used.

We look forward to receiving your revised manuscript.

Kind regards,

RunGuo Zang

Academic Editor

PLOS ONE

Additional Editor Comments:

The manuscript is generally well writtten and of interest to many ecologists.The manuscript should be acceptable after a major revision on basis of the concerns of the referee.

"The research was supported by the Slovak Grant Agency VEGA (project no. 1/0639/17), and by

392 project no. 20-06065S granted by the Grant Agency of the Czech Republic. We are grateful to

393 Natuurmonumenten (NL) for technical support and permission to carry out the fieldwork in their

394 nature reserves, and Frits Mohren for logistic support. We also thank Jan Willem Jongepier for

395 language revision, and reviewers for their comments."

Reviewers' comments:

Reviewer's Responses to Questions

**Comments to the Author**

1. Is the manuscript technically sound, and do the data support the conclusions?

Reviewer #1: Partly

2. Has the statistical analysis been performed appropriately and rigorously? 

Reviewer #1: No

3. Have the authors made all data underlying the findings in their manuscript fully available?

Reviewer #1: No

4. Is the manuscript presented in an intelligible fashion and written in standard English?

Reviewer #1: Yes

5. Review Comments to the Author

Reviewer #1: # General comments

In this manuscript, entitled “Two centuries of forest succession, and 30 years of vegetation changes in permanent plots in an inland sand dune area, The Netherlands”, the authors aim to describe long-term successional changes in an inland sand dune landscape situated in the Netherlands. The main asset of the article is to benefit from a long-term monitoring of the same dune environment in order to reconstruct the succession dynamics. I therefore think that the information provided by these articles is of interest to the scientific community, especially because they concern very particular environments. Nevertheless, I think that the manuscript still has important limitations that need to be clarified before it can be made acceptable for publication.

My main point of criticism would relate to the nature of the data, as well as the way it is interpreted and analyzed. To my opinion, these points deserve more context, justification and discussion. The data studied was indeed obtained using a synchronic approach (time since the beginning of the succession is estimated with the age of the trees), a diachronic approach (the same plots are studied at different periods) and finally a combination of the two (estimated age of the plot + time since the first sampling). The age of some sub-plot is furthermore interpolated according to the vegetation patch in which it is located (lines 145-146). In addition, only plot C is really studied over several years, where only the 1988 inventories for the plots A and B are considered.

I understand very well the difficulty of carrying out long-term ecological monitoring while avoiding any anthropogenic disturbance, so I don't think that this is sufficient to justify a rejection. Nevertheless, this raises important questions about the validity of the results because it means that for plot C, we have repeated measurements, whereas this is not the case for plots A and B. Further, the age of many subplots depends on a spatial unit of vegetation types, implying a dependence between the subplot situated in a same vegetation patch. This mixture of different data types, with different levels of independence, seems to me to be statistically “risky”, but the authors never mention this problem. Overall, the age of each subplot seems to be considered as an independent data, even though it corresponds to a plot sampled twice or if the age depends on a specific spatial unit. More generally, the diachronic and synchronic approaches are each subject to criticism (a combined diachronic/synchronic approach would therefore be all the more so) but this is a point that is generally not discussed. I would therefore encourage the authors to better justify their approach, or even modify their analyses to better take into account the inconsistent independence of the data. (Please note that this is why I noted “No” for question “Has the statistical analysis been performed appropriately and rigorously?” of the PLOS ONE Review Report, as an intermediate answer was not available)

More generally, I find that the structure of the article is sometimes relatively unclear. The introduction does not seem to highlight well the questions addressed by the authors while the discussion does not focus enough on the results (see specific comments for more details). In terms of data availability, I also disagree with the authors' statement that all data used in this study are available in the Supplementary Material. S1 presents data at the plot level, but not at the subplot level (which can vary greatly in age). It is therefore not possible to reproduce all analyses with the data currently available.

For these reasons, I believe that acceptance of the manuscript is still conditional on major corrections. I therefore encourage authors to revise their manuscript, hoping that my comments will be useful to them.

# Specific comments

Lines 38-39: This sentence seems to have little connection with the rest of the paragraph.

Lines 49-51: “Sand dune system” is repeated twice in the same sentence, maybe it could be rephrased

Lines 53-55 and 68-79: These passages seem to talk about the same idea but are in a different paragraph. It would be better to group these ideas together in the same places

Lines 60-66: The beginning of this section is more appropriate for the methodology, while the second section would be more relevant to the discussion.

Lines 60 -78: It seems strange to me to focus the introduction on the studied area, as this limits the overall scope of this work by confining it to a specific context.

Lines 92: I think a map and photos of the study area would be welcome (photos are however optional but it could help the reader to have a clearer picture of the study sites).

Lines 121-122: This sentence is redundant

Lines 124-127: Why is there two methods to classify the cover (ordinal scale 1-9 and percentage)?

Lines 135-140: What is the typology used to define these geomorphological units and what are the thresholds? Is it a relatively subjective classification?

Lines 149-152: This point should be discussed as, indeed, tree age is not always a reliable indicator of forest succession.

Line 157: Define the acronym TCV

Lines 161-162: Why a logarithmic transformation is used here?

Line 171: Why a 20 x 20 m grid? The experimental design is based on 10 x 10m grid so it’s difficult to understand the change of scale here

Line 172 : Would it be possible to separate the plot C1988 and C2018 (i.e., considering as covariables A1988, B1988, C1988 and C2018)?

Lines 172-175: The responses curves are GAM if I refer to the legend for Figure 3. However, the characteristics of these models are not presented in the manuscript or in the supplementary material. This would be suitable because if I refer to the method, it is not a fitting that is only meant to be descriptive.

Lines 184-186: Why calculate these areas again? Because the previous measurements were not reliable?

Lines 197-199: All the correlations (r and p) with the DCA axes should be provided in the manuscript

Figure 1: The figure should be divided in panels (A and B), it will facilitate the reading of the results.

Figure 2: Would it be possible to color the points referring to the plot and the year of sampling? In this way it could be seen whether the fitting works despite the special characteristics of the data. Could the authors also give the details of the fitting?

Figure 3: The figure should be divided in panels (A, B and C). Major and minor ticks would be necessary as well as a legend instead of the name of the trees next to the curves.

Figure 4: Placing these figures vertically could give them more space in the article and make them more readable. It would also be a good idea to include a legend with the color codes. Overall, I'm not convinced by the colors, especially white to describe Deschampsia flexuosa; white is more a color for the absence of results. Overall, I would advise to review the color code to get something more harmonious (for example a light brown instead of white).

Lines 293-308: This part should be at the end of the discussion, as it is a more general opening. I would advise to start the discussion with more concrete results.

Lines 351-355: I think this part needs more detail to better explain the implication of the results

Lines 362: Here, the authors could provide some example of the abiotic factors

Line 368: I think a word is missing after “strong dominant”

Line 380: Do the authors speak here of "late successional forests" broadly or for sand dune environments?

6. PLOS authors have the option to publish the peer review history of their article (what does this mean?). If published, this will include your full peer review and any attached files.

Reviewer #1: No

---

## [Author Response · Author response to Decision Letter 0]

21 Mar 2021

PONE-D-20-36698

Two centuries of forest succession, and 30 years of vegetation changes in permanent plots in an inland sand dune area, The Netherlands

PLOS ONE

Dear Dr. Prach,

Thank you for submitting your manuscript to PLOS ONE. After careful consideration, we feel that it has merit but does not fully meet PLOS ONE’s publication criteria as it currently stands. Therefore, we invite you to submit a revised version of the manuscript that addresses the points raised during the review process.

The manuscript is generally well writtten and of interest to many ecologists.The manuscript should be acceptable after a major revision on basis of the concerns of the referee. 

RESPONSE: Thank you and the referee for the very helpful comments. We have done our best to improve the manuscript as suggested and believe it would be now ready for publication.

Karel Prach, on behalf of the co-authors

RESPONSE: We included this (P5L104).

"The research was supported by the Slovak Grant Agency VEGA (project no. 1/0639/17), and by project no. 20-06065S granted by the Grant Agency of the Czech Republic. We are grateful to Natuurmonumenten (NL) for technical support and permission to carry out the fieldwork in their

nature reserves, and Frits Mohren for logistic support. We also thank Jan Willem Jongepier for language revision, and reviewers for their comments."

RESPONSE: We removed these from Acknowledgements and included the funding information only in the Funding Statement. We adapted the text in Funding Statement as follows:

The research was supported by the Slovak Grant Agency VEGA (projects no. 1/0639/17 and 1/0624/21), and by the project no. 20-06065S granted by the Grant Agency of the Czech Republic. The funders had no role in study design, data collection and analysis, decision to publish, or preparation of the manuscript.

RESPONSE: Thank you.

Reviewers' comments:

Reviewer's Responses to Questions

Comments to the Author

Reviewer #1: # General comments

In this manuscript, entitled “Two centuries of forest succession, and 30 years of vegetation changes in permanent plots in an inland sand dune area, The Netherlands”, the authors aim to describe long-term successional changes in an inland sand dune landscape situated in the Netherlands. The main asset of the article is to benefit from a long-term monitoring of the same dune environment in order to reconstruct the succession dynamics. I therefore think that the information provided by these articles is of interest to the scientific community, especially because they concern very particular environments. Nevertheless, I think that the manuscript still has important limitations that need to be clarified before it can be made acceptable for publication.

My main point of criticism would relate to the nature of the data, as well as the way it is interpreted and analyzed. To my opinion, these points deserve more context, justification and discussion. The data studied was indeed obtained using a synchronic approach (time since the beginning of the succession is estimated with the age of the trees), a diachronic approach (the same plots are studied at different periods) and finally a combination of the two (estimated age of the plot + time since the first sampling). The age of some sub-plot is furthermore interpolated according to the vegetation patch in which it is located (lines 145-146). In addition, only plot C is really studied over several years, where only the 1988 inventories for the plots A and B are considered.

I understand very well the difficulty of carrying out long-term ecological monitoring while avoiding any anthropogenic disturbance, so I don't think that this is sufficient to justify a rejection. Nevertheless, this raises important questions about the validity of the results because it means that for plot C, we have repeated measurements, whereas this is not the case for plots A and B. Further, the age of many subplots depends on a spatial unit of vegetation types, implying a dependence between the subplot situated in a same vegetation patch. This mixture of different data types, with different levels of independence, seems to me to be statistically “risky”, but the authors never mention this problem. Overall, the age of each subplot seems to be considered as an independent data, even though it corresponds to a plot sampled twice or if the age depends on a specific spatial unit. More generally, the diachronic and synchronic approaches are each subject to criticism (a combined diachronic/synchronic approach would therefore be all the more so) but this is a point that is generally not discussed. I would therefore encourage the authors to better justify their approach, or even modify their analyses to better take into account the inconsistent independence of the data. (Please note that this is why I noted “No” for question “Has the statistical analysis been performed appropriately and rigorously?” of the PLOS ONE Review Report, as an intermediate answer was not available)

RESPONSE: We agree with the reviewer that the data exhibit different levels of independence. On the other hand, it was the only way to realistically describe this succession regarding large geomorphological variability and its grain. Thus, three, differently aged plots of 200×200m in size were established in 1988 of which two (A,B) were, in the meantime, disturbed by human activities. For the overall description of succession we exploited the earlier data from all three plots which it is not a statistical problem in the unconstrained ordination of DCA (Šmilauer and Lepš 2014). For the other analyses, we excluded the plots A and B to avoid the combination of assynchronic vs. diachronic nature of our observations, i.e. the problematic combination of space-for-time substitution and resampling of one of the plots. Instead CCA, which we originally used, we tested effects of environmental variables on vegetation composition using the linear RDA analysis because the lenght of gradient was shorter after the exclusion of the plots A and B. Permutations were restricted both to avoid spatial autocorrelation (by defining the spatial grid) and for identity of subplot. Moreover, P values were adjusted by the Holm correction.

More generally, I find that the structure of the article is sometimes relatively unclear. The introduction does not seem to highlight well the questions addressed by the authors while the discussion does not focus enough on the results (see specific comments for more details). In terms of data availability, I also disagree with the authors' statement that all data used in this study are available in the Supplementary Material. S1 presents data at the plot level, but not at the subplot level (which can vary greatly in age). It is therefore not possible to reproduce all analyses with the data currently available.

RESPONSE: We re-arranged Introduction and included some new text with an effort to focus it better. We include all primary data into a public repository Dryadafter a possible final acceptance of the manuscript and we indicated this in the manuscript (L179).

For these reasons, I believe that acceptance of the manuscript is still conditional on major corrections. I therefore encourage authors to revise their manuscript, hoping that my comments will be useful to them.

RESPONSE: Thanks again for the comments, they were very helpful.

# Specific comments

Lines 38-39: This sentence seems to have little connection with the rest of the paragraph. 

RESPONSE: We deleted this sentence.

Lines 49-51: “Sand dune system” is repeated twice in the same sentence, maybe it could be rephrased 

RESPONSE: Adapted accordingly.

Lines 53-55 and 68-79: These passages seem to talk about the same idea but are in a different paragraph. It would be better to group these ideas together in the same places 

RESPONSE: We moved the first part to Page 4 to the end of the next paragraph, hoping now the text is more fluent.

Lines 60-66: The beginning of this section is more appropriate for the methodology, while the second section would be more relevant to the discussion. 

RESPONSE: We moved the first part to Methods as suggested, and rearranged the rest.

Lines 60 -78: It seems strange to me to focus the introduction on the studied area, as this limits the overall scope of this work by confining it to a specific context. 

RESPONSE: We moved the part concerning the study area to Methods.

Lines 92: I think a map and photos of the study area would be welcome (photos are however optional but it could help the reader to have a clearer picture of the study sites). 

RESPONSE: We newly included three photos illustrating each of the three studied sand dune complexes.

Lines 121-122: This sentence is redundant 

RESPONSE: We agree and we deleted it (the same information was already provided).

Lines 124-127: Why is there two methods to classify the cover (ordinal scale 1-9 and percentage)? 

RESPONSE: It is usual approach in phytosociology that the cover of the whole layers are estimated in percentage and that of particular species in an ordinal scale.

Lines 135-140: What is the typology used to define these geomorphological units and what are the thresholds? Is it a relatively subjective classification? 

RESPONSE: We included the reference where the units are described

Lines 149-152: This point should be discussed as, indeed, tree age is not always a reliable indicator of forest succession. 

RESPONSE: We are aware of some limitations, e.g. the possible existence of false rings, however, the wood core analysis is the best possible method to estimate the forest age if we know it is the first generation of trees and no felling happened. In the case of Scots Pine it is easy to count the rings as they are well visible.

Line 157: Define the acronym TCV 

RESPONSE: Explained at P8L177

Lines 161-162: Why a logarithmic transformation is used here? 

RESPONSE: It is the usual option to decrease the influence of dominant species (Šmilauer and Lepš 2014).

Line 171: Why a 20 x 20 m grid? The experimental design is based on 10 x 10m grid so it’s difficult to understand the change of scale here 

RESPONSE: Corrected, it was our mistake (in fact, it was a grid of 20 lines and 20 columns of 10 x 10m sublopts). 

Line 172 : Would it be possible to separate the plot C1988 and C2018 (i.e., considering as covariables A1988, B1988, C1988 and C2018)?

RESPONSE: We excluded A1988 and B1988 from the analyses where they were problematic and used only the repeatedly sampled plot C.

Lines 172-175: The responses curves are GAM if I refer to the legend for Figure 3. However, the characteristics of these models are not presented in the manuscript or in the supplementary material. This would be suitable because if I refer to the method, it is not a fitting that is only meant to be descriptive.

RESPONSE: Details on Generalized Aditive Model (GAM) were added to Methods.

Lines 184-186: Why calculate these areas again? Because the previous measurements were not reliable? 

RESPONSE: We modified the previous, not very precise formulation.

Lines 197-199: All the correlations (r and p) with the DCA axes should be provided in the manuscript 

RESPONSE: We enlarged the text (P10L227-231 but not providing really all correlations because the DCA diagram primarily presents overall vegetation pattern and the correlations between environmental factors and the axis scores are only illustrative. Their importance is exactly evaluated by the subsequent RDA analysis.

Figure 1: The figure should be divided in panels (A and B), it will facilitate the reading of the results. 

RESPONSE: We would prefer the original layout which is frequently used option in such diagrams and enables a direct visual comparison of both graphs. Moreover, there is enough space in the upper-right corner of the main diagram. However, the diagram was slightly adapted, and morer species are shown now to better correspond to S1 Table.

Figure 2: Would it be possible to color the points referring to the plot and the year of sampling? In this way it could be seen whether the fitting works despite the special characteristics of the data. Could the authors also give the details of the fitting? 

RESPONSE: We did it as suggested and provided more details in Methods.

Figure 3: The figure should be divided in panels (A, B and C). Major and minor ticks would be necessary as well as a legend instead of the name of the trees next to the curves. 

RESPONSE: We adapted the figure as suggested – it is arranged vertically in two panels and full species names are provided in the figure caption.

Figure 4: Placing these figures vertically could give them more space in the article and make them more readable. It would also be a good idea to include a legend with the color codes. Overall, I'm not convinced by the colors, especially white to describe Deschampsia flexuosa; white is more a color for the absence of results. Overall, I would advise to review the color code to get something more harmonious (for example a light brown instead of white). 

RESPONSE: We adapted the figure as suggested.

Lines 293-308: This part should be at the end of the discussion, as it is a more general opening. I would advise to start the discussion with more concrete results. 

RESPONSE: We carefully considered this suggestion but if we move this part of text further, the links to the previous and following paragraphs would be interrupted. We are aware of a rather subjective view but we prefer to start Discussion with main findings, then details, and then generalizations.

Lines 351-355: I think this part needs more detail to better explain the implication of the results 

RESPONSE: We enlarged the text. ) (P18L389-392).

Lines 362: Here, the authors could provide some example of the abiotic factors 

RESPONSE: We added two distinctive factors, wind and substrate texture.

Line 368: I think a word is missing after “strong dominant” 

RESPONSE: We adapted the text (L389).

Line 380: Do the authors speak here of "late successional forests" broadly or for sand dune environments? 

RESPONSE: We think "late successional forests" in general, we adapted the text.

---

## [Editor Report · Decision Letter 1]

30 Mar 2021

Two centuries of forest succession, and 30 years of vegetation changes in permanent plots in an inland sand dune area, The Netherlands

PONE-D-20-36698R1

Dear Dr. Prach,

We’re pleased to inform you that your manuscript has been judged scientifically suitable for publication and will be formally accepted for publication once it meets all outstanding technical requirements.

Kind regards,

RunGuo Zang

Academic Editor

PLOS ONE
---

## [Editor Report · Acceptance letter]

16 Apr 2021

PONE-D-20-36698R1 

Two centuries of forest succession, and 30 years of vegetation changes in permanent plots in an inland sand dune area, The Netherlands 

Dear Dr. Prach:

I'm pleased to inform you that your manuscript has been deemed suitable for publication in PLOS ONE. Congratulations! Your manuscript is now with our production department. 

Kind regards, 

on behalf of

Professor RunGuo Zang 

Academic Editor

PLOS ONE